# Longitudinal Internal Validity of the Quality of Life after Brain Injury: Response Shift and Responsiveness

**DOI:** 10.3390/jcm12093197

**Published:** 2023-04-29

**Authors:** Marina Zeldovich, Stefanie Hahm, Isabelle Mueller, Ugne Krenz, Fabian Bockhop, Nicole von Steinbuechel

**Affiliations:** 1Institute of Medical Psychology and Medical Sociology, University Medical Center Göttingen, Waldweg 37A, 37073 Göttingen, Germany; marina.zeldovich@med.uni-goettingen.de (M.Z.); im2622@cumc.columbia.edu (I.M.); ugne.krenz@med.uni-goettingen.de (U.K.); fabian.bockhop@med.uni-goettingen.de (F.B.); 2Department Health & Prevention, Institute of Psychology, University of Greifswald, Robert-Blum-Str. 13, 17489 Greifswald, Germany; stefanie.hahm@uni-greifswald.de; 3Department of Psychiatry, Columbia University Medical Center, 622 West 168th Street, New York, NY 10032, USA

**Keywords:** traumatic brain injury, disease-specific health-related quality of life, QoLIBRI, responsiveness, response shift

## Abstract

The Quality of Life after Brain Injury (QoLIBRI) questionnaire was developed and validated to assess disease-specific health-related quality of life (HRQoL) in individuals after TBI. The present study aims to determine its longitudinal validity by assessing its responsiveness and response shift from 3 to 6 months post-injury. Analyses were based on data from the European longitudinal observational cohort Collaborative European NeuroTrauma Effectiveness Research in Traumatic Brain Injury study. A total of 1659 individuals recovering from TBI were included in the analyses. Response shift was assessed using longitudinal measurement invariance testing within the confirmatory factor analyses framework. Responsiveness was analyzed using linear regression models that compared changes in functional recovery as measured by the Glasgow Outcome Scale–Extended (GOSE) with changes in the QoLIBRI scales from 3 to 6 months post-injury. Longitudinal tests of measurement invariance and analyses of discrepancies in practical significance indicated the absence of response shift. Changes in functional recovery status from three to six months were significantly associated with the responsiveness of the QoLIBRI scales over the same time period. The QoLIBRI can be used in longitudinal studies and is responsive to changes in an individual’s functional recovery during the first 6 months after TBI.

## 1. Introduction

Health-related quality of life (HRQoL) refers to the assessment of an individual’s perceived mental, physical, and social well-being and the overall ability to perform daily life activities [1]. Ideally, HRQoL instruments provide standardized information about the status and extent of limitations in a patient’s subjective experience of a medical condition or its treatment [2], capturing valuable insight into self-perceived health status and recovery patterns [3]. Measures of HRQoL can be generic or disease-specific. While generic measures allow for comparisons between different conditions and the general population, disease-specific instruments are more sensitive to selected health conditions, more precise in assessing their symptoms [4], and allow for a better prognosis, therapy, and rehabilitation recommendation compared with generic HRQoL measures. Due to this gain in nuanced information, disease-specific HRQoL is a recommended outcome that should be assessed in complex, heterogeneous diseases, such as traumatic brain injury (TBI) [3,5].

TBI is a significant cause of death and disability worldwide [6,7,8]. A recent epidemiological study reported almost 1.5 million hospital discharges and over 56,000 deaths due to TBI in Europe within 1 year, with TBI accounting for 37% of all injury-related deaths [9]. Despite the critical impact TBI has on society, public health, patients, and their next of kin, its effect is often insufficiently recognized [10,11]. TBI is considered an essential factor for decreased life expectancy, especially in young people [7,9,12], and is associated with increased morbidity and long-term disabilities in surviving individuals [8,13,14]. It can lead to lifelong limitations in daily activities due to lasting interference with cognitive, emotional, and physical functioning [15,16,17], including the development of post-concussion symptoms [18], impairments in motor functioning [19], and increased prevalence of mental health conditions such as anxiety [20,21,22], depression [21,22,23], or posttraumatic stress disorder [24]. These persistent medical and psychological repercussions often affect the HRQoL of individuals after TBI and their families [25].

The Quality of Life after Brain Injury (QoLIBRI) [26,27] is a TBI-specific patient-reported outcome measure (PROM) validated in more than 20 languages [26,28,29,30]. Its reliability and validity have been previously described in cross-sectional studies ranging from 3 months to 15 years after TBI [27,29,31,32,33,34,35]. The QoLIBRI is able to detect finer nuances between specific patient groups than the SF-36 measuring generic HRQoL [36], which is an important advantage for its use in clinical settings [4,37].

To date, little is known about the longitudinal association of disease-specific HRQoL after TBI. Some studies suggest a somewhat paradoxical increase in HRQoL despite disability [38]. Approaches to explain such a phenomenon include psychological processes such as reappraisal and adaption to the challenge of facing limiting health conditions [39]. These observed changes in perception of HRQoL after an adverse health event can likely be attributed to the adjustment of an individual’s internalized judgment of satisfactory health and HRQoL, usually referred to as a response shift [39,40]. While a response shift in assessing one’s health after a severe illness can be considered a common consequence of successful coping or intervention, it can also influence the psychometric indices of instruments assessing HRQoL [41,42]. Therefore, it is important to assess whether long-term evaluations with HRQoL instruments, like the QoLIBRI, reflect a true change in TBI or are susceptible to a potential response shift.

Response shifts may arise as a result of (1) a change in the responding individual’s internal standards of measurement, such as a recalibration, (2) a change in the values of the individual, such as reprioritization, or (3) a redefinition of the measured construct by the responding individual, such as reconceptualization [43]. The absence of a response shift legitimates comparing instruments’ scores at different time points to evaluate the long-term impact of disease and the efficacy of available treatments. In the case of PROMs such as QoLIBRI, careful assessment of potential response shift is a critical but rarely applied element in establishing the longitudinal validity of the instrument.

Another central characteristic of PROMs is their ability to detect actual change over time, referred to as responsiveness. The exact definition of responsiveness varies in the literature between (1) the ability to detect change in general, (2) measuring clinically meaningful change, and (3) assessing real change in the measured concept [44]. The first definition focuses on assessing the magnitude of a potential treatment effect and creates a lack of information about the instruments’ ability to detect the absence of a treatment effect correctly [44]. The second and third definitions face methodological challenges due to an insufficient classification of the term “clinically important change” and a lack of feasibility to measure the real change in the individuals’ HRQoL due to a scarcity of appropriate instruments. This results in a substantial overlap in measurement procedures. In both cases, researchers usually infer changes in HRQoL from clinical variables, a patient’s perception, or a doctor’s evaluation of change. Therefore, the evaluation of the responsiveness of an instrument varies across the literature, with the significance of change being potentially susceptible to the evaluator’s subjective judgment [44,45].

The present study aims to evaluate the longitudinal internal validity of the QoLIBRI. The assessment is conducted in two steps:Evaluation of response shift from three to six months after TBI using the longitudinal measurement invariance testing approach;Assessment of responsiveness as the ability of the QoLIBRI to detect changes in the patient’s functional recovery status, as measured by the Glasgow Outcome Scale–Extended (GOSE) [46], 3 to 6 months after injury.

Evidence of the absence of response shifts and the presence of responsiveness would suggest that the QoLIBRI can be used for longitudinal assessment of disease-specific HRQoL after TBI.

## 2. Materials and Methods

The present study utilizes data (core data set 3.0) from the prospective, longitudinal, observational Collaborative European NeuroTrauma Effectiveness Research in Traumatic Brain Injury study (CENTER-TBI; EC grant 602150; clinicaltrials.gov NCT02210221) aimed to improve characterization and classification of TBI. Inclusion criteria were a clinical diagnosis of TBI, clinical indication for a CT scan, presentation within 24 h after injury, and informed consent which was obtained according to local and national requirements. To avoid confounding outcome assessments, individuals with severe preexisting neurological disorders were excluded from the study [47]. Overall, *N* = 4509 (99.9% civilians) eligible patients included in the core study were stratified into 3 strata based on the clinical care pathways: emergency room (ER; discharge after ER admission), ward (admission to a hospital ward), and intensive care unit (ICU; admission to the ICU). Details on core sample characteristics are described elsewhere [48].

The present study focused on individuals after TBI who were at least 16 years old and had a GOSE status of 3 or higher at 3 and 6 months after injury. Responsiveness analyses were conducted with individuals for whom the QoLIBRI total score and 6 scale scores could be calculated at 3 and 6 months after injury (Sample 1: *N* = 1659). Confirmatory factor analyses and measurement invariance analyses were conducted only with complete QoLIBRI item data at 3 and 6 months post-injury (Sample 2: *N* = 1390). For more details on study sample attrition, see Figure 1.

### 2.1. Measures

#### 2.1.1. Sociodemographic Data

Sociodemographic data used in the present study were collected at enrollment and included the age, sex, marital status, educational level, and employment status of the study participants.

#### 2.1.2. Injury-Related Data

The severity of TBI was assessed using the Glasgow Coma Scale (GCS) [49], with values of 13–15 indicating mild, 9–12 moderate, and 3–8 severe TBI. It is often accessed several times within the first 24 h after injury (at the scene of the accident, in the first hospital, if the patient did not arrive directly at the study hospital, at the ER of the study hospital, and by post-stabilization). The GCS score was centrally imputed using IMPACT methodology [50]: The first post-stabilization value was taken and, if absent, the next available value back in time towards values at the accident scene. The GCS score was combined with the presence of intracranial abnormalities (ICA) detected by CT. Individuals were grouped into 4 categories according to the following cut-offs: uncomplicated mild (GCS ≥ 13 without ICA), complicated mild (GCS ≥ 13 with ICA), moderate (9 ≤ GCS ≤ 12), and severe TBI (GCS ≤ 8).

Injury severity was measured with the Injury Severity Score (ISS) [51], identifying three of twelve mostly injured body regions. The ISS is calculated as a sum of squares of the three body regions with the highest score. The maximal score for the ISS is 75. If any of the regions are assigned a score of 6, the ISS is automatically set to 75. Other TBI-related factors included the cause of injury (fall, road traffic accident, violent/other), clinical care pathways (emergency room (ER), admission, intensive care unit (ICU)), and the length of the hospital stay (in days).

Functional recovery status after TBI was rated using the GOSE [46] at three and six months after injury. The GOSE scores range from 1 to 8, covering the following functional recovery status: 1 = death, 2 = vegetative state, 3–4 = severe disability, 5–6 = moderate disability, and 7–8 = good recovery. The GOSE score was computed as a composite score combining information from the interview or, if not available, from the postal questionnaire (GOSE-Q [52] completed either by individuals after TBI or caretakers) or based on interviewer ratings for survivors. Since the GOSE-Q cannot distinguish between a vegetative state (score of 2) and a lower severe disability (score of 3), both categories were collapsed into one. Only patients with values of three and above participated in our study. Based on changes in the recovery status between three and six months post-injury, individuals after TBI were divided into three groups: if recovery status had changed to better, they were assigned to the “improved” group. Those with unchanged recovery status were attributed to the “stable” group, and the remainder to the “worsened” group.

The QoLIBRI [26,27] was used to assess TBI-specific HRQoL. The instrument comprises 37 Likert-type scaled items with five response options (“Not at all”, “Slightly”, “Moderately”, “Quite”, and “Very”) forming six domains (Cognitive, Self, Autonomy & Daily Life, Social Relationships, Emotions, and Physical Problems). The total score can be calculated with more than one-third of the responses and scaled to vary from 0 (worst possible HRQoL) to 100 (best possible HRQoL).

### 2.2. Statistical Analyses

#### 2.2.1. Sample Characteristics

We tested if patients who responded to the QoLIBRI at 3 and 6 months post-injury assessment differed from other TBI patients within six months after injury concerning age, sex, and injury-related characteristics such as clinical care pathways, TBI, and injury severity. The Welch two-sample *t*-test was applied for metric variables and Pearson’s chi-squared tests or permutation-based chi-squared tests (*n* < 5 observations per cell; *N* = 5000 permutations) for categorical ones. The significance level was set to α = 0.05.

#### 2.2.2. Confirmatory Factor Analysis

An optimal instrument structure at 3- and 6-month assessments was tested via confirmatory factor analysis (CFA) with the robust weighted least squares estimator (WLSMV) using the lavaan-package [53] in R [54]. Model fit was assessed with the scaled chi-square statistics, comparative fit index (CFI), root mean square error of approximation (RMSEA) with a 90-percent confidence interval, and standardized root mean square residual (SRMR). The standard cut-offs for CFI (>0.95), RMSEA (<0.06 for an excellent and 0.05 < RMSEA < 0.10 for a mediocre fit), and SRMR (<0.08) [55,56,57], indicating good model fit, have not been yet validated for the WLSMV estimator. Therefore, we tested four concurrent factorial structures (with 1. one factor, 2. two correlated factors, one factor that underlies positively formulated items and one negatively, 3. six correlated factors corresponding to QoLIBRI scales, 4. six correlated factors, and one common factor of higher order) and compared all fit indices across models. For the scaled chi-square difference test, α-level was set to 0.05.

#### 2.2.3. Response Shift

Response shift of each QoLIBRI scale was determined by testing for measurement invariance across the two time points within the framework of CFA for ordinal variables [58]. The content interpretation was performed based on Oort (2005) [59].

First, we tested a configural model with the same number of latent factors and the same pattern of zero and non-zero loadings across two time points. Failure of the model would indicate that participants reconceptualized their understanding of the HRQoL construct by the follow-up, attributing items to other factors.

Next, a loading model, constraining loadings of the configural model to be equal across time points, was investigated. The scaled chi-square test (α = 0.05) was used for model comparison. A worsened model fit would mean a response shift due to reprioritization: by the follow-up, the importance of some items has changed for the construct estimation.

Then, in the threshold model, we constrained the loading model’s thresholds to be equal across time points. A worsened model fit would mean a response shift due to recalibration (i.e., a change in the interpretation of response options): even if a participant reports the same level of HRQoL, he or she would score differently at the second time point. Invariance on the threshold level would be sufficient to conclude that the observed differences in the score means are due to the differences in the latent factor means [60].

Finally, in the residual model, we additionally constrained residual variances to be equal across time points. In the case of invariance, one can conclude that the responses showed no response shift, and all observed differences in the score means, variances, and covariances came from the corresponding differences in the latent factors.

In all models, latent factors were allowed to freely covary across time points. Residuals were allowed to freely correlate with themselves but not with other residuals across time points. The model specification was based on the marker item approach implying selecting marker variables based on the smallest difference in loadings over time [58]. In case where the loading, threshold, or residual model is non-invariant, Liu et al. (2017) [58] suggest testing if violations of invariance have a practical significance. The estimated parameters of each invariance model can be applied to calculate probabilities of choosing a particular response category under the corresponding model. In the case that, for example, the loading model holds, but the threshold model does not, the practical significance of an invariance violation can be attached through the differences in estimated response probabilities under the loading and threshold model for each item and all time points. Differences not exceeding 5% were considered negligible.

#### 2.2.4. Responsiveness

Responsiveness of an instrument reflects the extent to which changes in the measure relate to corresponding changes in an external reference measure over a defined time course. Regression analyses have proven useful in assessing responsiveness, as the regression coefficient (*b*) provides an easily interpretable index, and a goodness-of-fit assessment can be employed to check for the plausibility of the model [45]. Therefore, to evaluate the responsiveness of the QoLIBRI, linear regressions were calculated to find associations between the change in the QoLIBRI total and scale scores from three to six months post-injury and the change in the GOSE score from three to six months, using the R-package ‘lme4’ [61]. The GOSE change was included as an ordinal variable using two orthogonal contrasts to detect linear and non-linear (i.e., quadratic) associations with the QoLIBRI scores. Age, sex, marital status, education level, employment status, TBI severity, ISS, cause of injury, clinical care pathways, and length of hospital stay were used as covariates. Multiple imputations with the R-package ‘mice’ [62] was used to handle missing covariate data. The level of significance was set to α = 0.05 for the total score and was Bonferroni-corrected for the scale scores (α = 0.05/6 = 0.008).

The reliability of change scores was evaluated by receiver operating characteristic (ROC) curves with the R-package ’pROC’ [63] containing information on the sensitivity and specificity of the QoLIBRI change scores by discrimination between stable and improved patients. The area under the curve (AUC) was calculated; its values can vary between 0.5 and 1.0, with 1.0 indicating perfect discrimination and 0.5 discrimination not better than by chance [64].

Finally, we compared QoLIBRI total and scale scores between three and six months post-TBI using the Wilcoxon signed rank test for dependent samples to provide an overview of changes in the reported scores. The effect size *r* was calculated using the ‘wlicox_effsize’ function from the R package ‘rstatix’ [65]. Interpretation of values was based on conventional cut-offs: 0.10 ≤ *r* < 0.30 (small effect), 0.30 ≤ *r* < 0.50 (medium effect), and *r* ≥ 0.50 (large effect) [66]. Visualization was performed using strip plots showing trajectories of changes in HRQoL from three to six months.

## 3. Results

### 3.1. Study Participants

A total of 1659 patients (64.9% male) with a mean age of 49.61 years (*SD* = 19.15) were included in the present study (Sample 1, see Figure 1). The majority (approx. 77%) were either admitted to a hospital ward or an ICU after injury, while the remaining 23% were discharged after visiting the ER. Based on the GCS and information on intracranial abnormalities on the CT scans, 32.9% of the sample were diagnosed with uncomplicated mild TBI (i.e., GCS ≥ 13 and no abnormalities), 30.2% with complicated TBI (i.e., GCS ≥ 13 and visible intracranial abnormalities), 6.8% with moderate, and 14.2% with severe TBI; 6 months after TBI, 13.6% of participants showed a worsened functional state as rated by the GOSE, while for 55.3%, a stable, and for 31.1%, an improved functional status was observed. The patients’ mean ISS score was 18.04 (*SD* = 14.85). Participants included in the analysis sample differed significantly from those not included in all characteristics except for age. For more details, see Table 1. For the sample characteristics of the reduced sample (Sample 2, see Figure 1) as well as for the comparison between included and excluded individuals, see Appendix B, Table A1.

### 3.2. Confirmatory Factor Analysis

Table 2 provides the CFA results for 3 and 6 months post-injury, respectively. At both time points, model fit indices remarkably improved by the 6-factor and second-order models with CFIs above 0.949 and RMSEA values under 0.069. The scaled chi-square test identified that the 6-factorial structure fits best the QoLIBRI data at time points 3 (*χ*^2^(614) = 4121.54, *p* < 0.001, CFI = 0.954, RMSEA = 0.064, CI_90%_ [0.062; 0.066]) and 6 months post-injury (*χ*^2^(614) = 3775.89, *p* < 0.001, CFI = 0.963, RMSEA = 0.061, CI_90%_ [0.059; 0.063]). Therefore, this model was used for further analyses.

### 3.3. Response Shift

Table 3 presents the results from the longitudinal measurement invariance test for the QoLIBRI scales from three to six months after TBI. For each scale, we estimated four invariance models. In all cases, the CFI was high (≥0.971), and the SRMR did not exceed the cut-off of 0.06. Both indices showed minimal variation across invariance models for each scale, suggesting an adequate fit. However, the RMSEA of the scales Self (configural and loading model) and Social Relationships (configural model) was slightly increased (>0.10), indicating a worse model fit.

The scaled chi-square difference test indicated for all scales that adding loading invariance constraints did not significantly worsen the model fit when compared to the configural baseline model. For three of the six scales (Daily Life & Autonomy, Social Relationships, and Physical Problems), the scaled chi-square difference test indicated that the threshold model fit does significantly worsen the data in comparison to the loading invariance model. As the threshold invariance model did not hold up, an analysis of practical significance was conducted for all three scales [58]. Predicted probabilities for each of the scales showed only minimal differences between invariance models cross-sectionally and longitudinally. Analysis of the practical significance of invariance violation identified that the discrepancies in the estimated probabilities to choose a particular response option under concurrent models (e.g., loading and threshold invariance models for the Social Relationships scale) showed absolute differences not exceeding 2% (see Appendix A, Practical Significance–Discrepancies between Invariance Models). According to Liu and colleagues [58], small discrepancies in the predicted probabilities (<0.05) can be neglected as they only represent relatively few individuals. Therefore, we can assume that violations from threshold invariance for the Daily Life & Autonomy, Social Relationships, and Physical Problems scales were not caused by a response shift from three to six months post injury, and threshold invariance can be assumed.

For the remaining scales (Cognition, Self, and Emotions), the threshold invariance model held up compared to the loading invariance model. Consequently, the residual (unique factor) invariance model was tested. Scaled chi-square difference analysis indicated non-invariant residuals for all three scales. Again, an analysis of the practical significance of the failure of the residual invariance model was conducted. Predicted probabilities for each of the scales showed only minimal differences from three to six months (see Appendix A, Practical Significance–Discrepancies between Invariance Models). Consequently, their contribution to the latent construct HRQoL assessed by the QoLIBRI remained unchanged between the two measurement occasions, and observed differences in the scores over time can be attributed to true changes in the HRQoL.

The most pronounced changes between three and six months were observed on the Daily Life & Autonomy scale, where the participants became more likely to choose a higher response category. For instance, the predicted probability of choosing the answer “very” to the question “How satisfied are you with your ability to carry out domestic activities?” increased from 0.41 to 0.47 in the loading invariance model. The predicted probability for the question “How satisfied are you with the extent of your independence from others?” changed from 0.36 to 0.42 from 3 to 6 months in the response category “very” in the loading invariance mode. Furthermore, the item “How satisfied are you with your ability to get out and about” showed an increase in the endorsement of the higher response category from 0.46 to 0.52. For details, see Appendix A, Practical significance (Daily Life & Autonomy). These two questions represent the largest deviations in predicted probabilities for all scales over time, indicating that the scale may be more sensitive to measure change.

### 3.4. Responsiveness

According to the change in GOSE score between 3 and 6 months after TBI, recovery improved in 31.1% (*n* = 516) of our sample, 55.3% (*n* = 918) were classified as stable, and 13.6% (*n* = 225) had worsened recovery status. Change in the GOSE score, considered as a linear effect, contributed significantly to the change in the QoLIBRI total score, *B* = 4.129, *t*(1637) = 6.073, *p* < 0.001. At the scale level, a significant effect of GOSE change was observed for all scales except the Emotions scale. The proportion of variance in the change of the QoLIBRI score explained by the change in the GOSE scores was 4% for the total score and varied from 1% (Emotions scale) to 6% (Daily Life & Autonomy scale). Results showed no significant influence of the change in the GOSE score, considered as a quadratic effect. For details, see Table 4 and Appendix B, Table A2.

The average scores for the change in the QoLIBRI in relation to the change in the GOSE scores are shown in Figure 2. The most pronounced increase in the QoLIBRI score in relation to the GOSE score was observed in the Daily Life & Autonomy scale, followed by the Physical scale for those with improved functional recovery. For those with a worsened recovery status, the greatest decrease in the QoLIBRI scores was found in the Cognition and Self scales.

Additional ROC analysis showed that based on the QoLIBRI change scores, 60.4% of the individuals after TBI were correctly classified as presenting an improved vs. a not improved (i.e., stable or worsened) recovery status. The cut-off for the QoLIBRI change score, which maximizes sensitivity (55.4%) and specificity (63.9%) by distinguishing between improved and non-improved patients, was 2.5. For details, see Figure 3. The correctness of patients’ classification for scales ranged from 55.1% (Social Relationships) to 62.7% (Daily Life & Autonomy). The results of ROC analyses on the scale level are shown in Appendix B, Figure A1.

We observed a significant difference in the QoLIBRI total score from 3 to 6 months after TBI (*V* = 518408, *p* < 0.001, *r* = 0.12 corresponding to a small effect). The effect was mainly driven by the difference in the Daily Life & Autonomy and Physical Problems scales, both showing significant improvement from 3 to 6 months (*p* < 0.001). For visualization, see Figure 4 (QoLIBRI total score) and Figure A2 (for the scale scores).

## 4. Discussion

The present study aimed to assess the longitudinal internal validity of the QoLIBRI measuring TBI-specific HRQoL. This is the first study to assess response shift and responsiveness of the QoLIBRI for individuals between 3 and 6 months post TBI. This time frame is critical for patients as recovery mostly occurs in the first six months. For example, Gardner et al. [67] found that the majority of individuals after TBI achieve good to moderate recovery within the first half year after injury, with 70.9% following a gradual trajectory between the time points. Other studies also suggest that at least moderate recovery is reached within six months across all TBI severity groups [68,69].

However, most research has focused on functional recovery, thus neglecting domains that are additionally relevant to better understanding patients’ needs and facilitating the recuperation process [35]. In this context, especially the administration of PROMs (e.g., HRQoL measurement) can be considered as a comprehensive, economical, and reliable source of information complementary to the GOSE. Assessing both potential response shifts and responsiveness is critical to learning about actual changes over time when using PROMs longitudinally. In the present study, the absence of response shift and the demonstrated responsiveness of the QoLIBRI to the GOSE-assessed recovery status suggest that the instrument is useful for follow-up assessments during at least the first six months after injury. Some further aspects will be discussed in the following paragraphs.

### 4.1. Response Shift

For three QoLIBRI scales (i.e., Daily Life & Autonomy, Social Relationships, and Physical Problems), the longitudinal loading invariance model was attained, indicating that changes over time in the expected means measured by these QoLIBRI scales are entirely attributable to changes in the common latent factors over time [58]. Thus, the latent construct “HRQoL” estimated by the QoLIBRI remained unchanged between two measurement occasions, and observed differences in the scores over time can be attributed to true changes in the HRQoL. The other three scales (i.e., Cognition, Self, and Emotions) reached threshold invariance, pointing out that the observed differences in the score means are due to the differences in the latent factor means [60].

To better understand the discrepancies in response behavior from three to six months, a sensitivity analysis of the practical significance of the failure of invariance was conducted for all scales. The predicted probabilities revealed minor changes in response categories from three to six months post-TBI, with the largest variations on the Daily Life & Autonomy scale, where participants tended to endorse higher response categories indicating better HRQoL at six months post-TBI compared to three months. This may be explained by the progressive recovery process increasing their satisfaction with the autonomy they have gained during recovery. In our study, items measuring satisfaction with the level of ability to perform domestic activities, the level of independence from others, and the ability to get out and about showed increased endorsement six months after TBI. These findings are supported by previously published research suggesting considerable improvement in daily living activities during the first year post-injury [70].

In addition, discrepancies in the predicted probabilities between the retained and rejected measurement invariance models were calculated for each scale. However, none of the discrepancies between invariance models exceeded the 5% threshold, indicating that these differences can be neglected and the assumption of longitudinal measurement invariance can be retained.

The absence of response shift may also be explained by our study sample characteristics, as most of our patients had experienced a mild TBI. The intensive recovery processes for mild TBI occurs within the first weeks/months after injury [71]. In our sample, almost 60% of the individuals after TBI presented a good recovery at 3 months post-injury, which is comparable to other studies [67,68]. Moreover, patients after moderate and severe TBI who participated in the study might have felt generally better already when entering the study than those who did not take part. Therefore, adaptation and coping processes, which generally cause response shifts [43], might not have been pronounced in the individuals in our study sample.

### 4.2. Responsiveness

Linear regression modeling indicated that the linear change in the GOSE from three to six months was significantly associated with the change in HRQoL. This is in line with a recent study showing that the QoLIBRI was one of the most sensitive instruments for recovery status at 3 different time points (3, 6, and 12 months after TBI) across different patient groups [35]. Furthermore, some other studies have shown a significant association between unfavorable recovery and reduced (TBI-specific) HRQoL [29,48,72,73,74].

The only exception among the QoLIBRI scales was the change in the Emotions scale, which showed no significant association with the change in recovery status after Bonferroni adjustment for multiple testing. This may be explained by the fact that the GOSE focuses rather on functional (dis)ability, independence, social and leisure activities, and return to normal life after TBI and neglects the emotional status. As emotional well-being is crucial for the healing and recovery process and improvement of HRQoL in individuals after TBI [75], its assessment and, if necessary, treatment are mandatory in individuals after TBI. However, recent rehabilitation studies indicate a lack of services and treatment for post-TBI mental health conditions at all levels of severity [76,77].

Overall, the QoLIBRI and its scales appear to be sensitive to positive and negative changes in the participants’ HRQoL. Additional ROC analysis indicated that a QoLIBRI change score of 2.5 or higher could indicate significantly improved functional status and vice versa. Overall, a QoLIBRI change score of 2.5 correctly identified a significant change in recovery status in 60% of participants.

Finally, since the QoLIBRI can be considered responsive to changes in recovery from TBI and since we can assume that it measures true changes in TBI-specific HRQoL longitudinally (i.e., in a time frame of six months after injury), we can conclude that HRQoL improves significantly between three and six months, especially in terms of autonomy in daily living and physical problems.

### 4.3. Strengths and Limitations

To our knowledge, this is the first study systematically analyzing response shifts and responsiveness of the QoLIBRI and its scales. The main advantage is the large sample size, which also reflects the epidemiological distribution of the TBI severity in the general TBI population, allowing us to draw reliable conclusions. Some limitations should nevertheless be mentioned. Although advantageous for comparing with the general TBI population, the uneven distribution of TBI severity may pose some problems. The relatively small number of individuals after moderate and severe TBI does not allow for additional investigation of response shifts and responsiveness in these groups. Considering that higher TBI severity may be associated with a more pronounced decrease in TBI-specific HRQoL [78], additional analyses within the severe TBI group would be beneficial to gain more insight into potential changes in their HRQoL over time and the ability of the QoLIBRI to capture them. Furthermore, as participants included in the sample differed significantly from those not included concerning all characteristics except for age, the results should be interpreted with caution. Finally, the proportion of variance explained (4%) in the change in QoLIBRI score by the change in GOSE suggests that other factors not considered in the present study also contribute to changes in HRQoL. Therefore, the results should not be overinterpreted.

Future research using instruments other than QoLIBRI to assess TBI-specific HRQoL (e.g., TBI-QOL [79]) and data from other studies for external validation should be conducted to provide further evidence of the longitudinal validity of the QoLIBRI. Furthermore, the inclusion of additional time points after TBI would be beneficial to gain more insight into the variability of changes in TBI-specific HRQoL. Due to the design of the CENTER-TBI study, individuals after TBI who were admitted to an ER and subsequently discharged were only included in follow-up analyses up to six months after injury. Therefore, we were unable to perform analyses with this substantial group (i.e., 23%) beyond this time frame, which would have introduced sample bias due to the overrepresentation of severe or complex cases [80]. Thus, we decided to limit our analyses to the first six months after TBI. In addition, analyses of responsiveness to other relevant clinical comorbid conditions (e.g., depression, anxiety, post-traumatic stress disorder, post-concussion symptoms) may provide a more complete picture of how changes in TBI-specific HRQoL are related to outcomes other than post-TBI functional recovery. First, studies on the simultaneous consideration of these outcome domains point to their relevant impact on TBI-specific HRQoL [35,74]. Additional evidence of recovery using objective approaches such as CT and/or MRI, which were not available in the present study sample, would be beneficial for a more accurate, externally validated assessment of recovery.

## 5. Conclusions

Given the long-term impact of TBI on an individual’s life, and the heterogeneous pathways to symptom resolution and its potential negative impact on HRQoL, it is crucial to monitor outcomes such as HRQoL over time and not to rely solely on changes in functional status (i.e., the GOSE). To evaluate developmental trends straightforward, longitudinal validity, including response shift and responsiveness of the instruments, had to be established. Our results indicate the QoLIBRI can detect a true change in the underlying HRQoL construct and is sensitive to detecting changes in functional recovery status. The QoLIBRI, therefore, can be considered a valuable instrument to gain nuanced insight into the longitudinal development of recovery patterns and self-perceived health status in patients affected by TBI.

## Figures and Tables

**Figure 1 jcm-12-03197-f001:**
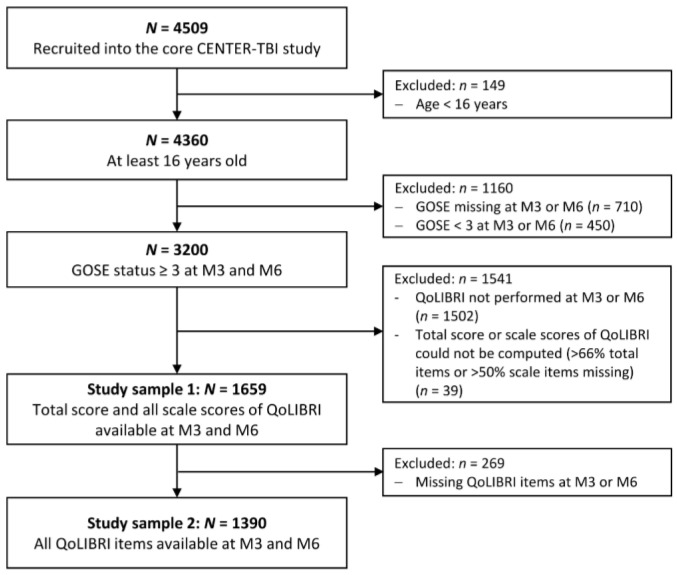
Sample attrition plot. Note. M3 = 3 months after TBI, M6 = 6 months after TBI, GOSE = Glasgow Outcome Scale–Extended.

**Figure 2 jcm-12-03197-f002:**
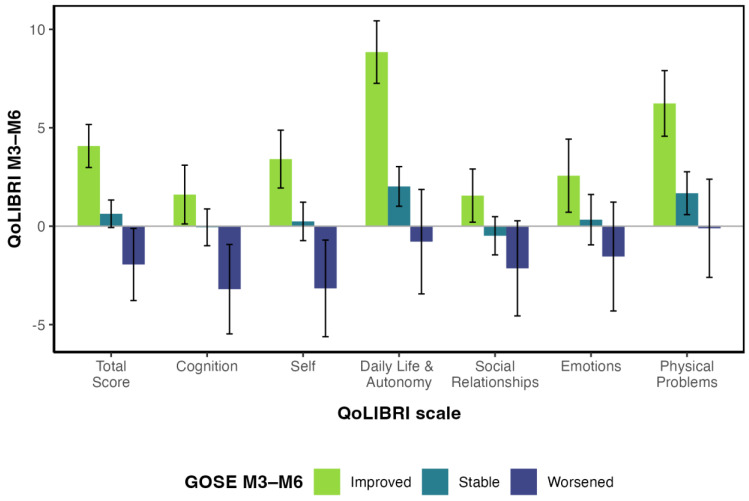
QoLIBRI mean change scores with 95% confidence intervals presented by changes in recovery status (GOSE). Note. M3 = three months after TBI, M6 = six months after TBI.

**Figure 3 jcm-12-03197-f003:**
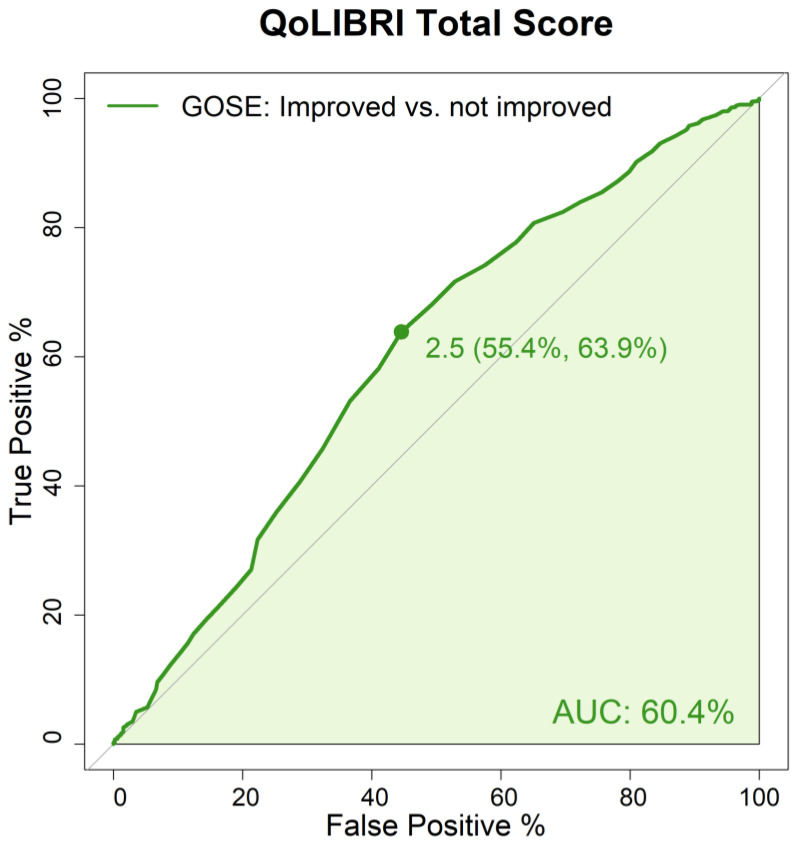
ROC analysis result for the QoLIBRI total score change in relation to the change in the GOSE score (improved vs. not improved).

**Figure 4 jcm-12-03197-f004:**
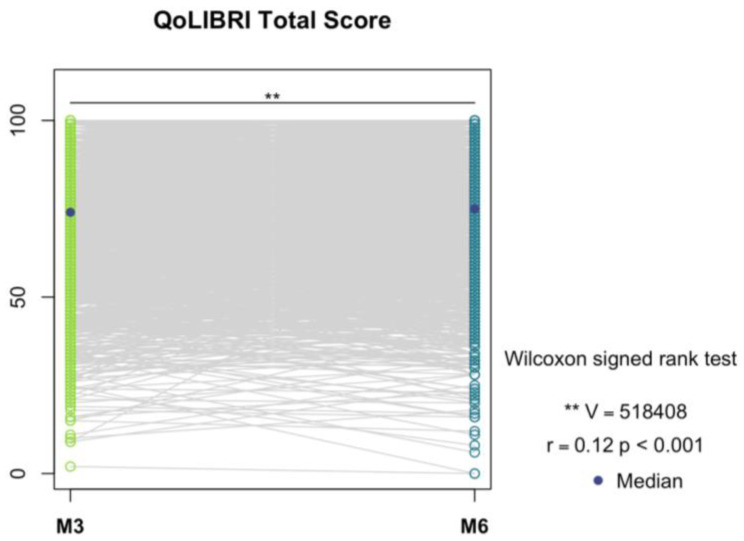
Differences in the QoLIBRI total score from three (M3) to six (M6) months after TBI. Note. ** significant at α = 0.05, *V* = Wilcoxon test statistic, *r* = effect size with the following classification: 0.10 ≤ *r* < 0.30 (small effect), 0.30 ≤ *r* < 0.50 (medium effect), and *r* ≥ 0.50 (large effect), *p* = *p*-value.

**Table 1 jcm-12-03197-t001:** Characteristics of the study sample included and not included in the analysis.

		Included in the Study Sample		
	Value/Group	Yes (*n* = 1659)	No (*n* = 2850)	*p*
Age	*M* (*SD*)	49.61 (19.15)	48.59 (22.45)	0.219
Sex	Male	1077 (64.92%)	1946 (68.28%)	0.022
	Female	582 (35.08%)	904 (31.72%)	
Marital status	Partnered	889 (53.59%)	1181 (41.44%)	<0.001
	Single	767 (46.23%)	1647 (57.79%)	
	Missing	3 (0.18%)	22 (0.77%)	
Education level	At least secondary/high school	547 (32.97%)	832 (29.19%)	<0.001
	College/university	464 (27.97%)	386 (13.54%)	
	None/primary school	207 (12.48%)	434 (15.23%)	
	Post-high school training	293 (17.66%)	403 (14.14%)	
	Missing	148 (8.92%)	795 (27.90%)	
Employment status	Full-time employed	718 (43.28%)	867 (30.42%)	<0.001
	In training	154 (9.28%)	332 (11.65%)	
	Part-time employed	196 (11.82%)	195 (6.84%)	
	Retired	389 (23.45%)	723 (25.37%)	
	Unemployed	123 (7.41%)	283 (9.93%)	
	Missing	79 (4.76%)	450 (15.79%)	
TBI severity (Glasgow Coma Scale; GCS)	Uncomplicated mild	546 (32.91%)	729 (25.58%)	<0.001
	Complicated mild	501 (30.20%)	563 (19.75%)	
	Moderate	113 (6.81%)	276 (9.68%)	
	Severe	236 (14.23%)	750 (26.32%)	
	Missing	263 (15.85%)	532 (18.67%)	
Injury severity score (ISS)	*M* (*SD*)	18.04 (14.85)	22.72 (17.79)	<0.001
Functional status change (Glasgow Outcome Scale–Extended; GOSE)	Improved	516 (31.10%)	494 (17.33%)	<0.001
	Stable	918 (55.34%)	1521 (53.37%)	
	Worsened	225 (13.56%)	113 (3.97%)	
	Missing	0 (0%)	722 (25.33%)	
Cause of injury	Fall	715 (43.10%)	1309 (45.93%)	<0.001
	Road traffic accident	692 (41.71%))	990 (34.74%)	
	Violent/other	251 (15.13%)	532 (18.67%)	
	Missing	1 (0.06%)	19 (0.66%)	
Clinical care pathways	ER	383 (23.09%)	465 (16.32%)	<0.001
	Admission	640 (38.58%)	883 (30.98%)	
	ICU	636 (38.34%)	1502 (52.70%)	
Length of hospital stay (in days)	*M* (*SD*)	10.56 (18.18)	13.30 (22.60)	<0.001
	Missing	29 (1.75%)	89 (3.12%)	

Note. *n* = absolute frequencies, *p* = *p*-value, *M* = mean, *SD* = Standard deviation. Welch’s *t*-test was used for all continuous variables due to non-normal distribution; *χ*^2^-tests and permutation-based *χ*^2^-tests (*n* < 5 observations per cell; *N* = 5000 permutations) were used for categorical data.

**Table 2 jcm-12-03197-t002:** Results of the confirmatory factor analysis (*N* = 1390).

	Confirmatory Factor Analysis				*χ*^2^ Difference Test	
Model	*χ* ^2^	CFI	RMSEA [CI_90%_]	SRMR	∆*χ*^2^ (∆df)	*p*
3 months post-injury						
One Factor	15,281.52 (629)	0.821	0.130 [0.128; 0.131]	0.096	-	-
Two Factor	10,939.70 (628)	0.874	0.109 [0.107; 0.111]	0.078	460.77 (1)	<0.001
**Six Factor**	**4121.54 (614)**	**0.954**	**0.064 [0.062; 0.066]**	**0.041**	**1766.03 (14)**	**<0.001**
Two Level	4532.72 (623)	0.949	0.067 [0.065; 0.069]	0.052	251.92 (9)	<0.001
6 months post-injury						
One Factor	14,024.43 (629)	0.842	0.124 [0.122; 0.126]	0.086	-	-
Two Factor	10,358.82 (628)	0.885	0.106 [0.104; 0.107]	0.070	414.20 (1)	<0.001
**Six Factor**	**3775.89 (614)**	**0.963**	**0.061 [0.059; 0.063]**	**0.037**	**1854.56 (14)**	**<0.001**
Two Level	4153.73 (623)	0.958	0.064 [0.062; 0.066]	0.047	229.28 (9)	<0.001

Note. CFI = comparative fit index (>0.95), RMSEA = root mean square error of approximation (<0.06) with a 90-percent confidence interval, SRMR = standardized root mean square residual (<0.08). Model in **bold** shows the best fit.

**Table 3 jcm-12-03197-t003:** Results of the measurement invariance analysis (*N* = 1390).

	Measurement Invariance Analysis				*χ*^2^ Difference Test	
Model	*χ*^2^ (*df*)	CFI	RSMEA [CI_90%_]	SRMR	∆*χ*^2^ (∆df)	*p*
Cognition						
Configural	594.04 (69)	0.988	0.074 [0.069; 0.080]	0.027	-	-
Loading	598.08 (75)	0.988	0.071 [0.066; 0.076]	0.027	3.34 (6)	0.766
Threshold	577.46 (95)	0.989	0.060 [0.056; 0.065]	0.027	14.50 (20)	0.804
Residual	545.55 (102)	0.990	0.056 [0.051; 0.061]	0.028	27.26 (7)	*<0.001*
Self						
Configural	1357.32 (69)	0.971	0.116 [0.111; 0.121]	0.039	-	-
Loading	1372.96 (75)	0.971	0.112 [0.106; 0.117]	0.039	5.63 (6)	0.466
Threshold	1397.01 (95)	0.972	0.099 [0.095; 0.104]	0.039	31.11 (20)	0.054
Residual	1260.64 (102)	0.980	0.090 [0.086; 0.095]	0.040	20.41 (7)	*0.005*
Daily Life & Autonomy						
Configural	558.81 (69)	0.988	0.071 [0.066; 0.077]	0.027	-	-
Loading	565.64 (75)	0.988	0.069 [0.063; 0.074]	0.027	6.69 (6)	0.350
Threshold	592.84 (95)	0.989	0.061 [0.057; 0.066]	0.027	33.41 (20)	*0.030*
Residual *	-	-	-	-	-	-
Social Relationships						
Configural	718.35 (47)	0.971	0.101 [0.095; 0.108]	0.039	-	-
Loading	736.49 (52)	0.971	0.097 [0.091; 0.104]	0.039	7.06 (5)	0.216
Threshold	728.63 (68)	0.972	0.084 [0.078; 0.089]	0.039	31.13 (16)	*0.010*
Residual *	-	-	-	-	-	*-*
Emotions						
Configural	196.23 (29)	0.988	0.064 [0.056; 0.073]	0.024	-	-
Loading	198.94 (33)	0.988	0.060 [0.052; 0.068]	0.024	3.74 (4)	0.443
Threshold	195.40 (47)	0.989	0.048 [0.041; 0.055]	0.024	13.02 (14)	0.525
Residual	200.39 (52)	0.990	0.045 [0.039; 0.052]	0.025	15.10 (5)	*0.010*
Physical Problems						
Configural	228.77 (29)	0.988	0.070 [0.062; 0.079]	0.035	-	-
Loading	228.48 (33)	0.988	0.065 [0.057; 0.073]	0.034	3.86 (4)	0.426
Threshold	259.25 (47)	0.989	0.057 [0.050; 0.064]	0.035	30.04 (14)	*0.008*
Residual *	-	-	-	-	-	-

* Residual invariance for Daily Life & Autonomy, Social Relationships, and Physical problems scales not reported; practical significance examined instead (see Appendix A). Note. *χ*^2^ = chi-square value, *df* = degrees of freedom, CFI = comparative fit index (>0.95), RMSEA = root mean square error of approximation (<0.06) with a 90-percent confidence interval, SRMR = standardized root mean square residual (<0.08). Values in *italic* are significant at 5%.

**Table 4 jcm-12-03197-t004:** Results of regression analyses.

Scale	Predictor	*B* (SE)	*b*	*t*	*p*
Total	GOSE (linear)	4.129 (0.680)	0.344	6.073	**<0.001**
	GOSE (quadratic)	0.223 (0.511)	0.019	0.436	0.663
Cognition *	GOSE (linear)	3.365 (0.904)	0.213	3.722	**<0.001**
	GOSE (quadratic)	−0.758 (0.680)	−0.048	−1.114	0.265
Self *	GOSE (linear)	4.725 (0.932)	0.289	5.070	**<0.001**
	GOSE (quadratic)	−0.223 (0.701)	−0.014	−0.318	0.750
Daily Life & Autonomy *	GOSE (linear)	6.384 (0.977)	0.365	6.536	**<0.001**
	GOSE (quadratic)	1.204 (0.735)	0.069	1.639	0.102
Social Relationships *	GOSE (linear)	2.646 (0.900)	0.168	2.942	**0.003**
	GOSE (quadratic)	0.035 (0.676)	0.002	0.051	0.959
Emotions *	GOSE (linear)	2.985 (1.177)	0.145	2.536	0.011
	GOSE (quadratic)	0.259 (0.886)	0.013	0.293	0.770
Physical Problems *	GOSE (linear)	3.960 (1.024)	0.219	3.866	**<0.001**
	GOSE (quadratic)	1.207 (0.771)	0.067	1.565	0.118

Note. Results of regression analyses controlling for the following variables: age, sex, marital status, education level, employment status, TBI severity, injury severity (ISS), cause of injury, clinical care pathways, length of hospital stay (for details, see Appendix B-Table A2). *B* = unstandardized regression coefficient, SE = standard error, *b* = standardized regression coefficient, *t* = *t*-value, *p* = *p*-value. Values in **bold** are significant at 5% (total score) or at 0.8% (scale scores; Bonferroni-adjusted marked with *).

## Data Availability

All relevant data are available upon request from CENTER-TBI, and the authors are not legally allowed to share it publicly. The authors confirm that they received no special access privileges to the data. CENTER-TBI is committed to data sharing and, in particular, to responsible further use of the data. Hereto, we have a data sharing statement in place: https://www.center-tbi.eu/data/sharing (accessed on 27 April 2023). The CENTER-TBI Management Committee, in collaboration with the General Assembly, established the Data Sharing policy and Publication and Authorship Guidelines to ensure the correct and appropriate use of the data as the dataset is hugely complex and requires the help of experts from the Data Curation Team or Bio-Statistical Team for correct use. This means that we encourage researchers to contact the CENTER-TBI team for any research plans and the Data Curation Team for any help in the appropriate use of the data, including sharing of scripts. Requests for data access can be submitted online: https://www.center-tbi.eu/data (accessed on 27 April 2023). The complete Manual for data access is also available online: https://www.center-tbi.eu/files/SOP-Manual-DAPR-2402020.pdf (accessed on 27 April 2023).

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
