# Peer review of "Longitudinal Internal Validity of the Quality of Life after Brain Injury: Response Shift and Responsiveness"

_jcm, 2023, doi:10.3390/jcm12093197_

Round 1

Reviewer 1 Report

This study aims to assess the longitudinal validity of the QoLIBRI 3 to 6 months after TBI.  The participants for this study came from the CENTER-TBI cohort and included patients with TBI without preexisting conditions. Authors assessed the QoLIBRI on this cohort and found it to be a valid tool in assessing outcomes at 3 and 6 months post-tBI. This is a well-designed and written study that deserves inclusion in the literature. 

Author Response

Response: Dear Reviewer, we would like to thank you very much for your very encouraging feedback.

Reviewer 2 Report

Thank you for submitting your manuscript “Longitudinal validity of the Quality of Life After Brain Injury: response shift and responsiveness”.  The paper is overall well written. Overall the study is an analysis of QoLIBRI questionnaire as a sub analysis of the larger dataset from the European longitudinal observational cohort Collaborative European Neurotrauma Effectiveness Research in TBI study.  My comments are largely general in relation to the larger study:

1.      Introduction:  The introduction should be shortened as the QoLIBRI is well described in the literature and the overall manuscript is lengthy. Recommend citing more and less reiteration of citations

2.     Materials and Methods:  Similar recommendation here, less QoLIBRI description and focus on the interval methods.

3.     Statistical analysis:  The analyses supports internal validation and not external validation and that is not clear from the title through to the analysis section

4.     Discussion:  Section is confusing as it reads partly as a methods paper and partly as a clinical effectiveness paper.  In effect, it reads as there are measurable changes between 3-6 months and also comments on the significances clinically and methodologically on the differences identified.  Agree with the inherent limitations of the study and the final conclusion regarding more data for its utility at time points beyond 6months.  This may require comparisons to longer time frame scales and therefore the manuscript is largely applicable to 3 and 6 months.

In essence, the paper could be significantly shortened since the QoLIBRI is already well published.  The clinical utility of 3 vs 6 months should be strengthened or the paper be more written as a methods paper. Beyond 6 months is difficult to comment on considering this is a sub analysis with data limited data set and time constraints.  

Author Response

Thank you for submitting your manuscript “Longitudinal validity of the Quality of Life After Brain Injury: response shift and responsiveness”.  The paper is overall well written. Overall the study is an analysis of QoLIBRI questionnaire as a sub analysis of the larger dataset from the European longitudinal observational cohort Collaborative European Neurotrauma Effectiveness Research in TBI study.  My comments are largely general in relation to the larger study:

Response: Dear Reviewer, we would like to thank you for your valuable feedback. Below, you find our detailed responses. 

  1. Introduction:  The introduction should be shortened as the QoLIBRI is well described in the literature and the overall manuscript is lengthy. Recommend citing more and less reiteration of citations

Response: Thank you for this comment. We have shortened the introduction while trying to retain the information necessary to explain the rationale for this study.

P. 2, lines 64 ff.:

“The Quality of Life after Brain Injury (QoLIBRI) [26,27] is a TBI-specific pa-tient-reported outcome measure (PROM) validated in more than 20 languages [26,28–30]. Its reliability and validity have been previously described in cross-sectional studies ranging from three months to 15 years after TBI [27,31,31–34,29,35]. The QoLIBRI is able to detect finer nuances between specific patient groups than the SF-36 measuring generic HRQoL [36], which is an important advantage for its use in clinical settings [4,37].”

  1. Materials and Methods:  Similar recommendation here, less QoLIBRI description and focus on the interval methods.

Response: We have shortened the description of the QoLIBRI in the Introduction section and left only a short paragraph in the Methods section containing information skipped in the Introduction.

P. 5, lines 237 ff.:

“The QoLIBRI [26,27] was used to assess TBI-specific HRQoL. The instrument comprises 37 Likert-type scaled items with five response options (“Not at all”, “Slightly”, “Moderately”, “Quite”, “Very”) forming six domains (Cognitive, Self, Autonomy & Daily Life, Social Relationships, Emotions, and Physical Problems). The total score can be calculated with more than one-third of the responses and scaled to vary from 0 (worst possible HRQoL) to 100 (best possible HRQoL).”

  1. Statistical analysis: The analyses supports internal validation and not external validation and that is not clear from the title through to the analysis section

Response: Thank you for raising this important issue. Indeed, external validation additionally using instruments other than QoLIBRI to assess TBI-specific HRQoL (e.g., TBI-QOL) or data other than those from the CENTER-TBI should be performed to provide further evidence of the longitudinal validity of the QoLIBRI. We have added information on the kind of validity in the title and study aims and provided a comment in the Discussion section.

Adjusted title: “Longitudinal internal validity of the Quality of Life After Brain Injury: response shift and responsiveness”.

Study aims: “The present study aims to evaluate the longitudinal internal validity of the QoLIBRI”.

P. 16, lines 616 ff.:

“Future research additionally using instruments other than QoLIBRI to assess TBI-specific HRQoL (e.g., TBI-QOL [79]) and data from other studies for external validation should be conducted to provide further evidence of the longitudinal validity of the QoLIBRI.”

  1. Discussion:  Section is confusing as it reads partly as a methods paper and partly as a clinical effectiveness paper. In effect, it reads as there are measurable changes between 3-6 months and also comments on the significances clinically and methodologically on the differences identified.  Agree with the inherent limitations of the study and the final conclusion regarding more data for its utility at time points beyond 6months.  This may require comparisons to longer time frame scales and therefore the manuscript is largely applicable to 3 and 6 months.

Response: Thank you for this comment. We have reorganized and rewritten some parts of the discussion and tried to focus more on the clinical implications by translating the methodological findings into a clinical context. We have emphasized the importance of the 3-6 month assessments and the need to assess disease-specific HRQoL in patients after TBI with regard to a longer time frame.

P. 14, lines 474 ff.:

“The present study aimed to assess the longitudinal internal validity of the QoLIBRI measuring TBI-specific HRQoL. This is the first study to assess response shift and responsiveness of the QoLIBRI for individuals between three and six months post TBI. This time frame is critical for patients as recovery mostly occurs in the first six months. For example, Gardner et al. [67] found that the majority of individuals after TBI achieve good to moderate recovery within the first half year after injury, with 70.9% following a gradual trajectory between the time points. Other studies also suggest that at least moderate recovery is reached by six months across all TBI severity groups [68,69].

However, most research has focused on functional recovery, thus neglecting domains which are additionally relevant to better understand patients’ needs and facilitate the recuperation process [35]. In this context, especially the administration of PROMs (e.g., HRQoL measurement) can be considered as a comprehensive, economical and reliable source of information complementary to the GOSE. Assessing both potential response shifts and responsiveness is critical to learning about actual changes over time when using PROMs longitudinally. In the present study, the absence response shift and the demonstrated responsiveness of the QoLIBRI to the GOSE-assessed recovery status suggest that the instrument is useful for follow-up assessments during at least the first six months after injury. Some further aspects will be discussed in the following paragraphs. Some further aspects will be discussed in the following paragraphs.”

We also added subsections 4.1. Response Shifts and 4.2. Responsiveness and removed unnecessary repetitions from the Methods/Results. We have also emphasized the need for validation of QoLIBRI for longitudinal assessments beyond the six-month time frame:

 P. 16, lines 618 ff.:

“Furthermore, the inclusion of additional time points after TBI would be beneficial to gain more insight into the variability of changes in TBI-specific HRQoL. Due to the design of the CENTER-TBI study, individuals after TBI who were admitted to an ER and subsequently discharged were only included in follow-up analyses up to six months after injury. Therefore, we were unable to perform analyses with this substantial group (i.e., 23%) beyond this time frame, which would have introduced sample bias due to the overrepresentation of severe or complex cases [80]. Thus, we decided to limit our analyses to the first six months after TBI.”

In essence, the paper could be significantly shortened since the QoLIBRI is already well published.  The clinical utility of 3 vs 6 months should be strengthened or the paper be more written as a methods paper. Beyond 6 months is difficult to comment on considering this is a sub analysis with data limited data set and time constraints. 

Response: Thank you very much! We did our best to follow your suggestions to improve the manuscript.

Reviewer 3 Report

The research article provides evidence that the Quality of Life after Brain Injury (QoLIBRI) questionnaire could be used in longitudinal studies of TBI individuals’ functional recovery during the first six months after the injury. The manuscript is well written, and here are some minor comments. 

Is any comorbidity, such as PTSD, Sleep, substance use, etc., observed in TBI patients? What measures were taken to correct them?

Are these samples only from civilians? Are there any samples included from the military, including blast-induced TBI?

On what basis was GOSE measured three months after injury? Any specific reason for not being measured below three months after the TBI?

Is there any difference in response shift and responsiveness among the different TBI grades?

Are any efforts to this subjective evidence correlated with objective evidence of recoveries such as a CT scan or MRI?

Author Response

The research article provides evidence that the Quality of Life after Brain Injury (QoLIBRI) questionnaire could be used in longitudinal studies of TBI individuals’ functional recovery during the first six months after the injury. The manuscript is well written, and here are some minor comments. 

Response: Dear Reviewer, we would like to thank you for your encouraging feedback. Below, you find our detailed responses. 

Is any comorbidity, such as PTSD, Sleep, substance use, etc., observed in TBI patients? What measures were taken to correct them?

Response: We observed clinically relevant anxiety, depression, and PTSD in patients, associated with reduced HRQoL and vice versa. We have already mentioned the simultaneous consideration of multiple outcomes in the Discussion section (see P. 16, L. 629 ff.). However, in the present study we focused on the longitudinal validity of the QoLIBRI without considering other outcomes.

P. 16, lines 626 ff.:

“In addition, analyses of responsiveness to other relevant clinical comorbid conditions (e.g., depression, anxiety, post-traumatic stress disorder, post-concussion symptoms) may provide a more complete picture of how changes in TBI-specific HRQoL are related to outcomes other than post-TBI functional recovery. First studies on the simultaneous consideration of these outcome domains point to their relevant impact on TBI-specific HRQoL [35,74].”

Are these samples only from civilians? Are there any samples included from the military, including blast-induced TBI?

Response: The CENTER TBI study focused on civilians. There are <0.1% patients in the core study (N = 4509) who sustained TBI due to military deployment. We have added a note on this sample characteristic in the Methods.

P. 3, lines 182 ff.:

“Overall, N = 4509 (99.9% civilians) eligible patients included in the core study were stratified into three strata based on the clinical care pathways: emergency room (ER; discharge after ER admission), ward (admission to a hospital ward), and intensive care unit (ICU; admission to the ICU).”

On what basis was GOSE measured three months after injury? Any specific reason for not being measured below three months after the TBI?

Response: All outcomes, including recovery status, were measured at multiple time points: Baseline, 2 weeks, 3 months, 6 months, 12 months, and 24 months to longitudinally capture patient state of health and recovery from TBI. We used 3-month and 6-month data to relate functional recovery to HRQoL and to test the ability of the QoLIBRI to capture changes in recovery, because the study design did not foresee data collection beyond six months after injury in patients who were initially admitted to the emergency department and then discharged home. We have added some limitations regarding this issue:

P. 16, L. 618 ff.:

“Furthermore, the inclusion of additional time points after TBI would be beneficial to gain more insight into the variability of changes in TBI-specific HRQoL. Due to the design of the CENTER-TBI study, individuals after TBI who were admitted to an ER and subsequently discharged were only included in follow-up analyses up to six months after injury. Therefore, we were unable to perform analyses with this substantial group (i.e., 23%) beyond this time frame, which would have introduced sample bias due to the overrepresentation of severe or complex cases [80]. Thus, we decided to limit our analyses to the first six months after TBI.”

Is there any difference in response shift and responsiveness among the different TBI grades?

Response: Thank you for raising this important issue. Unfortunately, due to the distribution of TBI severity in our sample, with a majority of mild cases, we are unable to answer this question or provide additional analyses. We have added some restrictions regarding this issue in the Limitations section. Below, please find the updated version:

P. 16, L. 604 ff.:

The relatively small number of individuals after moderate and severe TBI does not allow for additional investigation of response shift and responsiveness in these groups. Considering that higher TBI severity may be associated with a more pronounced decrease in TBI-specific HRQoL [78], additional analyses within the severe TBI group would be beneficial to gain more insight into potential changes in their HRQoL over time and the ability of the QoLIBRI to capture them.”

Are any efforts to this subjective evidence correlated with objective evidence of recoveries such as a CT scan or MRI?

Response: Patients underwent a CT on admission/study enrollment and had further scans only if clinically indicated. Therefore, we only have systematic CT information at baseline. We used this to differentiate between uncomplicated mild TBI (GCS ≥ 13 and no abnormalities detected on CT scan) and complicated mild TBI (GCS ≥ 13 and abnormalities detected on CT scan) (P. 4, L. 212 ff.). Regarding MRI, a sub-study recruited approximately 500 patients. These patients (mostly admitted to an ICU) underwent their first MRI within three weeks after TBI. We have added the following to the Limitations:

P. 16, L. 613 ff.:

“Additional evidence of recovery using objective approaches such as CT and/or MRI, which were not available in the present study sample, would be beneficial for a more accurate, externally validated assessment of recovery.”

Thank you!